# An Energy-Based Method for Lifetime Assessment on High-Strength-Steel Welded Joints under Different Pre-Strain Levels

**DOI:** 10.3390/ma15134558

**Published:** 2022-06-28

**Authors:** Chengji Mi, Zhonglin Huang, Haibo Wang, Dong Zhang, Tao Xiong, Haigen Jian, Jiachang Tang, Jianwu Yu

**Affiliations:** 1Department of Mechanical Engineering, Hunan University of Technology, Zhuzhou 412007, China; michengji_86@126.com (C.M.); happy2021hut@126.com (Z.H.); bug2021@126.com (D.Z.); 13838@hut.edu.cn (T.X.); 2Department of Mechanical Engineering, Hunan Automotive Engineering Vocational College, Zhuzhou 412001, China; hyfree69@163.com; 3Department of Mechanical and Vehicle Engineering, Hunan University, Changsha 410082, China; mcj20112011@hnu.edu.cn

**Keywords:** pre-strain, strain energy density, fatigue life estimation, welded joint, high-strength steel Q345

## Abstract

Pre-loading on engineering materials or structures may produce pre-strain, especially plastic strain, which would change the fatigue failure mechanism during their service time. In this paper, an energy-based method for fatigue life prediction on high-strength-steel welded joints under different pre-strain levels was presented. Tensile pre-strain at three pre-strain levels of 0.2%, 0.35% and 0.5% was performed on the specimens of the material Q345, and the cyclic stress and strain responses with pre-loading were compared with those without pre-loading at the same strain level. The experimental work showed that the plastic strain energy density of pre-strained welded joints was enlarged, while the elastic strain energy density of pre-strained welded joints was reduced. Then, based on the strain energy density method, a fatigue life estimation model of the high-strength-steel welded joints in consideration of pre-straining was proposed. The predicted results agreed well with the test data. Finally, the validity of the developed model was verified by the experimental data from TWIP steel Fe-18 Mn and complex-phase steel CP800.

## 1. Introduction

Fatigue fracture is one of the failure modes of mechanical structures, such as welded joints; according to a survey, over 80% of mechanical failure is caused by cyclic loadings [1]. In general, before the welded components are in service, the manufacturing and the assembling process could result in pre-strain on the welded structures, even plastic strain, which would contribute to fatigue damage in the future [2,3,4]. Therefore, taking the pre-straining effect into account is extremely important for accurately predicting the remaining life of welded joints.

It has been reported that pre-deformation has a great influence on the fatigue behavior of engineering materials and structures. From a microcosmic perspective, pre-loading could introduce unfavorable factors into the welded joints to shorten their in-service time in advance, such as twins, dislocations or other deformation defects. It has been said that face-centered cubic materials with low stacking fault energy are especially sensitive to pre-strain when it presents low-cycle fatigue behavior dominated by the planar slip mode [5]. For example, the planar slip is easily generated by the deformation behavior of austenitic stainless steel under cyclic loadings, and is related to the loading history [6]. Some researchers found that the fatigue life of 316 L stainless steel could be shortened through the irreversibility of the dislocation structure inherited by the pre-strain and variability of strain localization during the fatigue process [7]. However, some aluminum alloys, such as 7075-T6, are likely to produce cross-slip under cyclic loadings, indicating that the deformation behavior was independent of the pre-strain [8,9], while the life-span of the aluminum alloy 6061-T6 with compressive pre-strain would decrease in virtue of slip grains and permanent slip bands caused by the increase in the plastic strain amplitude under cyclic loading [10,11]. In fact, the material could be hardened by the pre-straining, so the strength can be improved to achieve better high-cycle fatigue performance. For example, the yield stress of dual-phase steel with 600 MPa strength increased after pre-straining, while its uniform elongation reduced. In addition, the low-cycle life expectancy decreased with an increase in pre-straining, and it increased in the high-cycle life span [12,13,14,15].

Due to the initial defects left by welding in welded structures, as well as stress concentration and welding residual stress, plastic deformation is mainly concentrated along the weld toe when a load is applied [16,17,18,19,20]. If there is pre-strain on the welded joints, the fatigue behavior of the weld seam will be more complicated [21]. Currently, researchers have mainly focused on how pre-strain affects the fatigue limit and crack growth of welded joints [22,23]. In order to predict the fatigue life of welded joints under different pre-strain paths, a strain–life model with the exponential mean stress correction term was suggested [24,25]. However, the plastic strain generated by pre-strain is usually non-uniformly distributed, and pre-strain may promote the growth of micro-cracks and micro-pores in advance, so it is extremely challenging to consider the strain as the fatigue damage parameter for predicting the life span of welded joints [26,27,28,29,30,31]. In this paper, an energy-based method for fatigue life prediction of high-strength-steel (Q345) welded joints under different pre-strain levels of 0.2%, 0.35% and 0.5% was proposed. The strain energy density was defined as the fatigue parameter based on the experimental work.

## 2. Experimental Work

### 2.1. Tensile Test

In this paper, the specimen was produced by arc welding, and the parental material was low-alloy high-strength steel Q345. The dimensions are shown in Figure 1. The total length of the specimen was 140 mm, and the length of the measuring region in the middle was 40 mm. The width was 8 mm and the thickness was 6 mm. The chemical composition of the steel is listed in Table 1. The specimen was grasped by the upper and lower clamping heads, as shown in Figure 2. For the monotonic tensile test, the tensile rate was 0.001 mm/s, and all tests were conducted at room temperature on account of testing standard GB/T228.1-2010. The stress–strain curve is shown in Figure 3. The mechanical parameters are listed in Table 2. The low-strength matching approach was utilized to connect the material, and so that the mechanical parameters in the welded area were slightly lower than those of the base metal.

### 2.2. Fatigue Test

The pre-strain was produced by applying uniaxial tension along the length direction under the strain-control mode. The strain rate had a speed of 2 × 10^−4^ s^−1^, and there were three strain levels, 0.2% (named PR-0.2%), 0.35% (named PR-0.35%) and 0.5% (named PR-0.5%). Then, the final residual stretching pre-strain on the specimen was 0.06%, 0.21% and 0.35%, respectively. The test without pre-strain was marked as AR. Then, based on the strain control, fatigue tests were conducted on the specimens at three strain amplitudes of 0.2%, 0.15% and 0.1%. All tests were carried out with a stress ratio of R = −1 according to the testing standard GB/T15248-2008. The loading frequency was from 0.5 Hz to 1 Hz.

## 3. Results and Discussion

### 3.1. Effect of Pre-Strain on Fatigue Behavior

The stress–strain data at different strain levels were obtained by an extensometer, and the cyclic stress–strain response curve of an AR specimen is shown in Figure 4. It can be seen that the peak stress of the welded specimen under cyclic loadings at the half cycle was significantly lower than that at the second cycle, which showed the cyclic softening of the material.

After pre-straining along the tensile direction, the hysteresis line of the welded joint appeared to be asymmetric at the beginning of the fatigue process, as shown in Figure 5. The peak tensile stress in the first cycle was higher than the peak compressive stress at the same strain amplitude. As a result, the average tensile stress was generated after pre-straining. However, the peak tensile stress decreased rapidly in the subsequent cycles, and the hysteresis line at 0.1 Nf was close to that at 0.5 Nf. The significant reduction of the average stress indicated that the hardening effect produced by the pre-strain was rapidly depleted after the cyclic loadings.

The cyclic stress–strain responses of the welded specimen under different pre-strains are compared in Figure 6. It can be seen that the peak stress at half cycle gradually decreased with the increase in the pre-strain levels, and the proportion of elastic strain amplitude in the total strain amplitude also decreased with the increase in pre-strain, which made the area of the hysteresis loop gradually enlarge under the tensile pre-strain. The reason for this phenomenon was due to the fact that deformation twins were generated during fatigue deformation, and the total number of twins was basically constant throughout the process. The formation of pre-strain-induced deformation twins reduced the number of twins generated during fatigue deformation [32,33]. This resulted in an increase in the proportion of the plastic part in the strain amplitude and also affected the fatigue properties of the material. The cyclic results at different strain amplitudes are shown in Figure 7. The welded specimen with higher pre-strain had a shorter life span, and it was generally believed that the decrease in the ductility of the material caused by the tensile pre-strain lessened the resistance to low-cycle fatigue [34]. The experimental work showed that the increase in pre-strain accelerated the cyclic softening of the welded specimen.

### 3.2. Fatigue Life Prediction of Welded Joints under Pre-Strain

#### 3.2.1. Strain Energy Density Method

The energy-based method uses the strain energy density as a parameter to measure fatigue damage, which is a scalar quantity and is a suitable fatigue parameter for predicting the fatigue behavior of high-strength-steel welded joints under pre-strain [27,28,29,30,31,32,33,34]. According to the literature of Chengji Mi [35], the total strain energy density for each cycle could be considered as a fatigue damage parameter for both low and high-cycle fatigue, which included the plastic strain energy density and the elastic strain energy density. In fully symmetric cyclic loading, the plastic strain energy density of the material was the area enclosed by the hysteresis loop at half cycle under cyclic loadings, as shown in Figure 8.

The stress–strain response encloses a completely closed hysteresis line, and its plastic strain energy density is expressed as [27,28,29,30,31,32,33,34,35]:(1)ΔWp=∮εpminεpmaxσdεp
where σ is the tensile stress, $\varepsilon_{p}$ is the plastic strain and $\varepsilon_{p}^{\max},\varepsilon_{p}^{\min}$ are the maximum and minimum plastic strain, respectively.

The tensile positive elastic strain energy is calculated as [27,28,29,30,31,32,33,34,35], assuming that the linear relation *σ* = *Eε* is valid in the elastic regime:(2)$$\Delta W_{e+}=\frac{\sigma_{\max}{}^{2}}{2E}$$

Combined with the definition of elastic strain energy in fatigue modeling [34,35], a rapidly decaying average stress is introduced to describe the effect generated by the pre-strain, and Equation (2) can be rewritten as:(3)$$\Delta W_{e+}=\frac{1}{2E}{\left(\frac{\Delta \sigma }{2}+\sigma_{m}\right)}^{2}$$
where $\sigma_{\max}$ is the maximum tensile stress value, E is the elastic modulus, Δσ is the cyclic stress range and $\sigma_{m}$ is the average stress.

The total strain energy density can be determined by summing the plastic strain energy and the tensile positive elastic strain energy density, and its relationship with fatigue life can be expressed by the following equation [27,28,29,30,31,32,33,34,35]:(4)$$\Delta W^{t}=\Delta W^{e+}+\Delta W^{p}=A{\left(2N_{f}\right)}^{B}$$
where $2N_{f}$ is the fatigue life, A is the strain energy density coefficient and B is the strain energy density index.

Based on the strain energy density method mentioned above and experimental data, the elastic, plastic and total strain energy densities of the welded specimen with and without pre-strain are listed in Table 3. The relationship between the total strain energy density and fatigue life is shown in Figure 9.

#### 3.2.2. Strain Energy Density Method under Pre-Strain

Based on the calculated results from Table 3, it can be seen that the total strain energy density at the same strain amplitude was close to a constant, with or without pre-strain, but the elastic strain energy density decreased with increases in pre-strain, and the plastic strain energy density had the opposite tendency. The effect of pre-strain on fatigue life in view of an energy damage mechanism was reflected by the variation in the elastic strain energy density and plastic strain energy density. According to the model proposed by Chengji Mi [35], the energy-based model can be derived as:(5)$$\Delta W^{t}=C_{1}{\left(2N_{f}\right)}^{d_{1}}+C_{2}{\left(2N_{f}\right)}^{d_{2}}$$
where $C_{1}$, $C_{2}$, $d_{1}$ and $d_{2}$ are relative with the material’s parameters.

The relationship between the elastic and plastic strain energy and pre-strain at 0.15% strain amplitude is shown in Figure 10. It can be seen that there was a kind of exponential relationship between them, which can be expressed by the following equation:(6)$$\Delta W^{e+}=k_{1}{\left(\varepsilon_{pr}\right)}^{n_{1}}+b_{1}$$
(7)$$\Delta W^{p}=k_{2}{\left(\varepsilon_{pr}\right)}^{n_{2}}+b_{2}$$

If the pre-strain $\varepsilon_{pr}$ is zero, the constants $b_{1}$ and $b_{2}$ stand for the elastic and plastic strain energy density, respectively, while $k_{1}$, $k_{2}$, $n_{1}$ and $n_{2}$ are material parameters.

Then the relationship between the elastic and plastic strain energy density with pre-strain can be rewritten as
(8)$$\Delta W_{pr}{}^{e+}=k_{1}{\left(\varepsilon_{pr}\right)}^{n_{1}}+\Delta W^{e+}$$
(9)$$\Delta W_{pr}{}^{p}=k_{2}{\left(\varepsilon_{pr}\right)}^{n_{2}}+\Delta W_{}{}^{p}$$

This relationship can be further simplified to be a power law form:(10)$$\Delta W_{pr}{}^{e+}={\left(1+\alpha_{1}\varepsilon_{pr}\right)}^{\beta_{1}}\Delta W^{e+}$$
(11)$$\Delta W_{pr}{}^{p}={\left(1+\alpha_{2}\varepsilon_{pr}\right)}^{\beta_{2}}\Delta W^{p}$$

The total strain energy density with pre-strain can be described as:(12)$$\Delta W^{t}=\Delta W_{pr}{}^{e+}+\Delta W_{pr}{}^{p}={\left(1+\alpha_{1}\varepsilon_{pr}\right)}^{\beta_{1}}C_{1}{\left(2N_{f}\right)}^{d_{1}}+{\left(1+\alpha_{2}\varepsilon_{pr}\right)}^{\beta_{2}}C_{2}{\left(2N_{f}\right)}^{d_{2}}$$
where $\Delta W_{pr}{}^{e+}$ and $\Delta W_{pr}{}^{p}$ are the elastic strain energy density and plastic strain energy density with pre-strain, respectively; and $\alpha_{1}$, $\beta_{1}$, $\alpha_{2}$ and $\beta_{2}$ are the material parameters of the welded joints.

The fatigue parameters of the welded specimen in Equation (12) were determined from the data in Figure 10, and are listed in Table 4. The relationship between the total strain energy density and fatigue life are shown in a double logarithmic coordinate system, as plotted in Figure 11. The fitting curves at different pre-strain levels could meet the experimental data.

To verify the applicability of the suggested model, the estimated life span of steel Fe-18 Mn with and without pre-strain based on Equation (12) was close to the experimental data [32], shown in Figure 12a. Figure 12b shows that the predicted lifetime of complex-phase steel CP800 with and without pre-strain based on the suggested model matched well with the tested data [36]. The black dashed lines are used to represent the scatter band of fatigue life with an error factor of two. All data were in the region between the two dashed lines. The results indicated that the model could describe the effect of the pre-strain on the fatigue properties of welded joints very well.

## 4. Conclusions

In this paper, the effect of pre-strain on the welded joints of high-strength steel was studied. Strain-controlled fatigue life tests were conducted on welded specimens with and without pre-strain. The experimental data showed that the tensile pre-strain resulted in a reduction in the ductility of the welded Q345 steel to weaken the resistance to low-cycle fatigue and accelerate the process of cyclic softening. The relationship between the elastic and plastic strain energy density of the welded joints with and without pre-strain and pre-strain levels was constructed. Then, a new strain energy density model considering the pre-strain effect was proposed to describe the fatigue performances of welded joints, and the validity of the developed model was verified by the experimental data from TWIP steel Fe-18 Mn and complex-phase steel CP800 [32,36].

## Figures and Tables

**Figure 1 materials-15-04558-f001:**
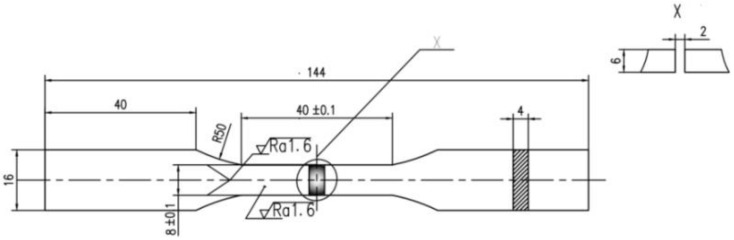
Dimensions (in mm) of the welded specimen.

**Figure 2 materials-15-04558-f002:**
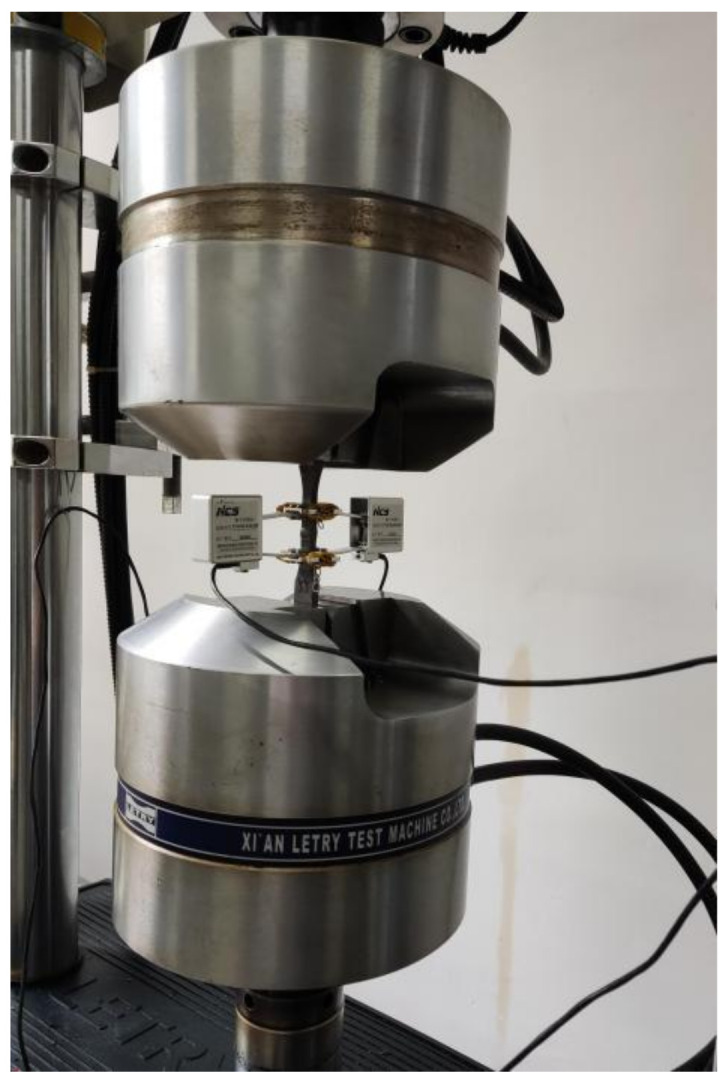
Clamping display of the specimen.

**Figure 3 materials-15-04558-f003:**
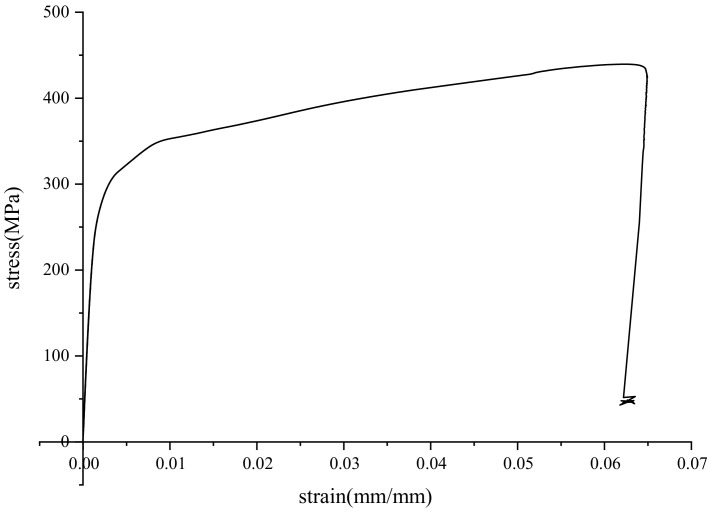
Stress–strain curve of the welded joint.

**Figure 4 materials-15-04558-f004:**
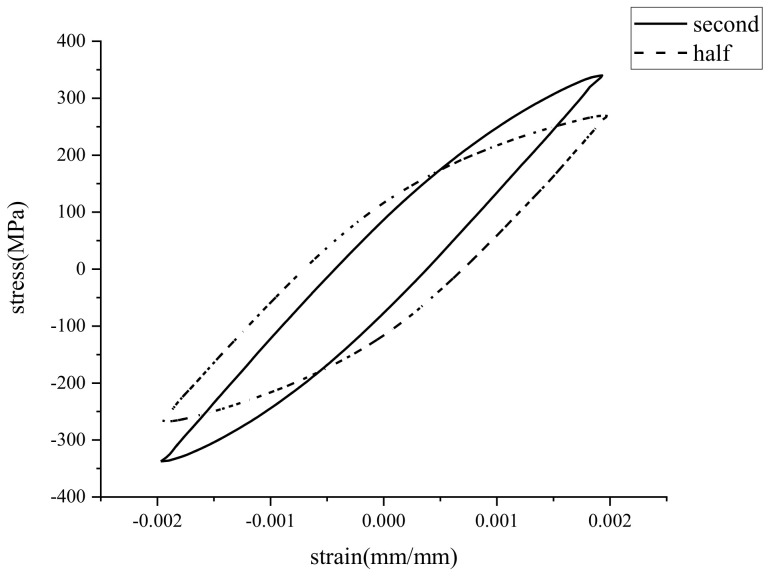
Cycle stress−strain curve for specimen.

**Figure 5 materials-15-04558-f005:**
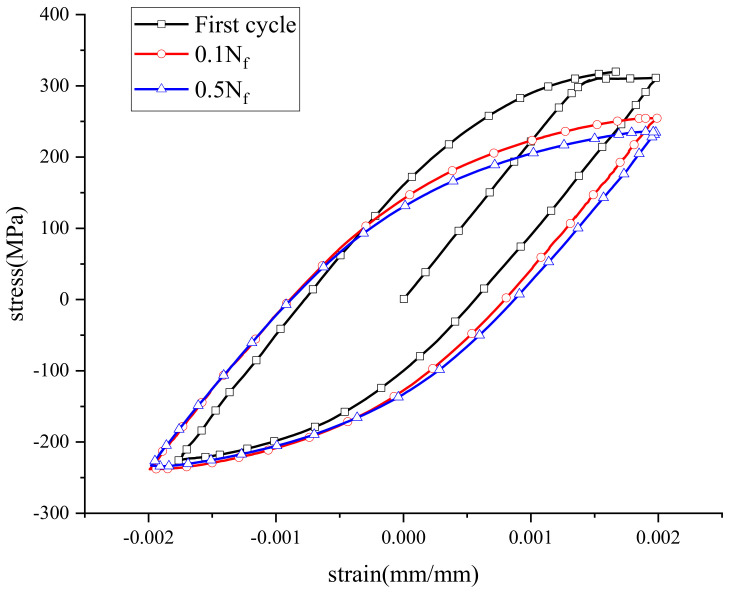
Effect of pre−strain on the hysteresis line.

**Figure 6 materials-15-04558-f006:**
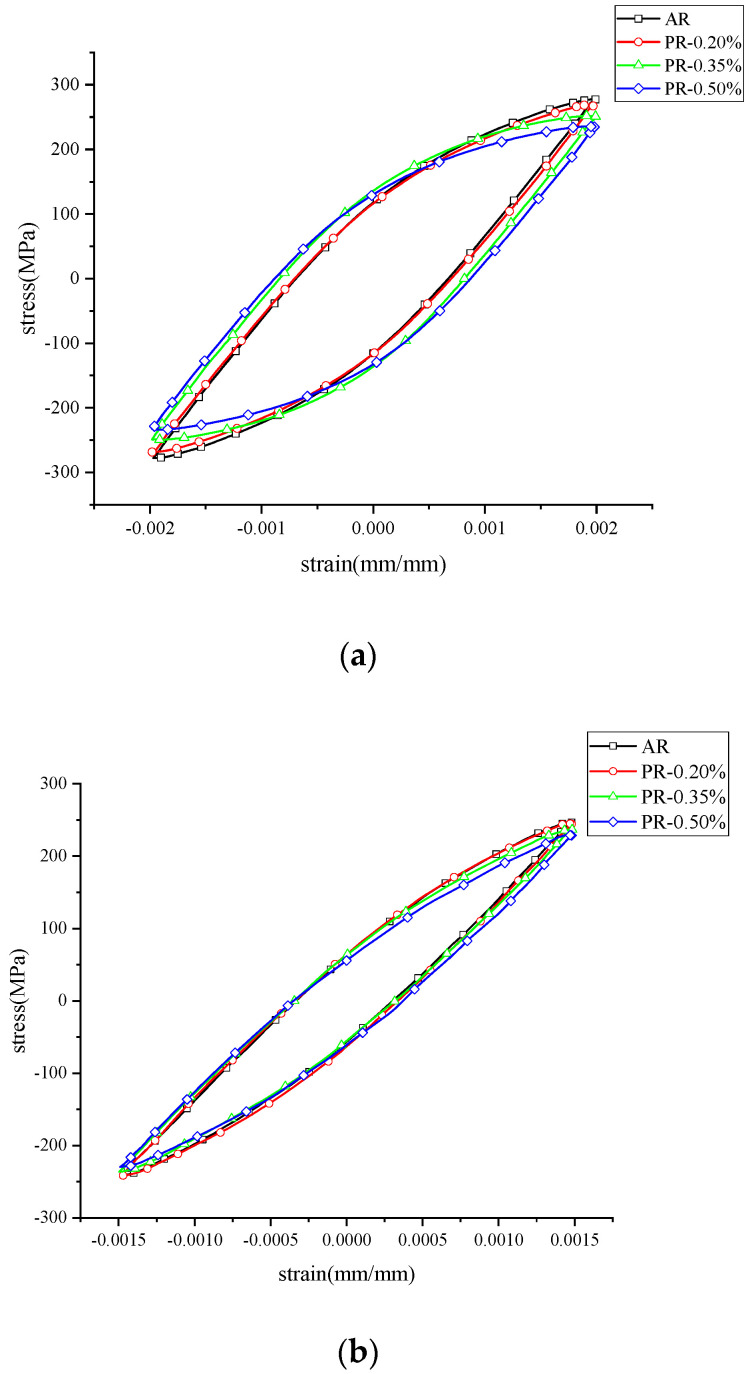
Half−cycle stressstrain response at different strain levels. (**a**) 0.2% strain amplitude; (**b**) 0.15% strain amplitude; (**c**) 0.1% strain amplitude.

**Figure 7 materials-15-04558-f007:**
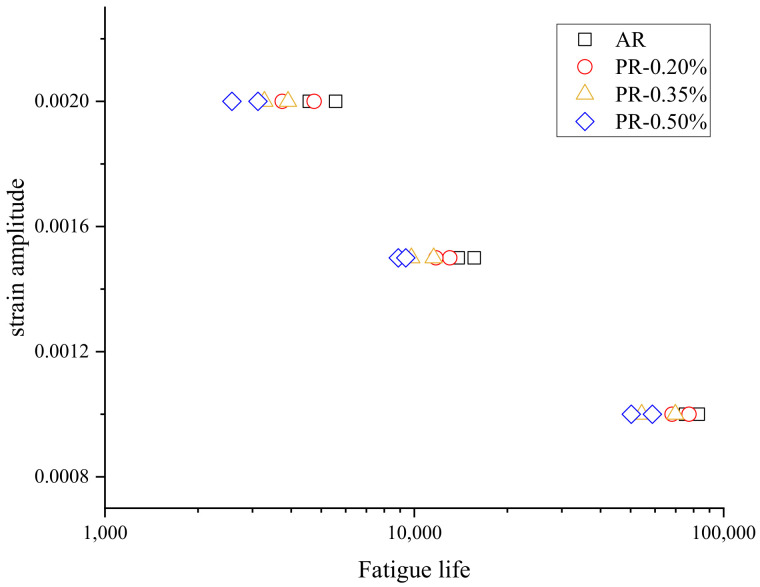
Strain amplitude fatigue life diagram.

**Figure 8 materials-15-04558-f008:**
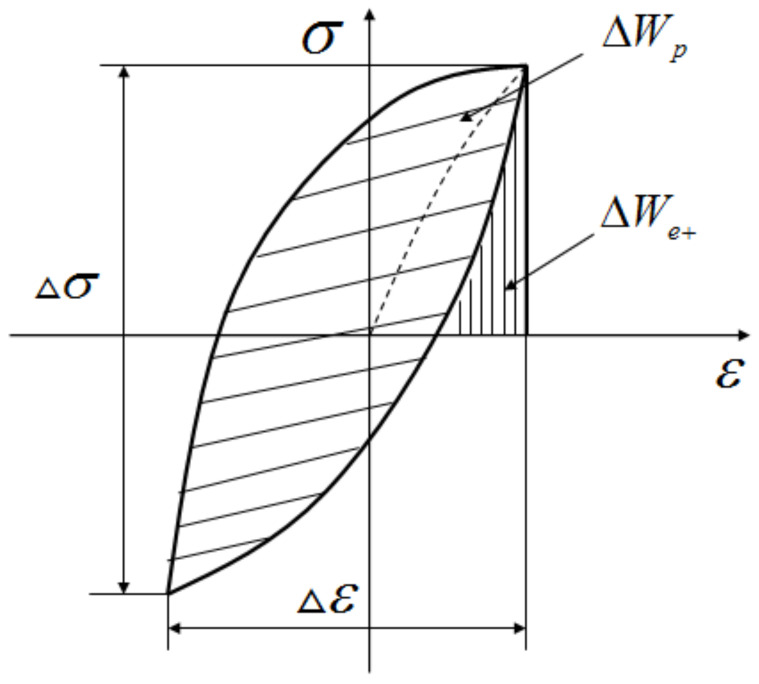
Calculation of elastic and plastic strain energy density.

**Figure 9 materials-15-04558-f009:**
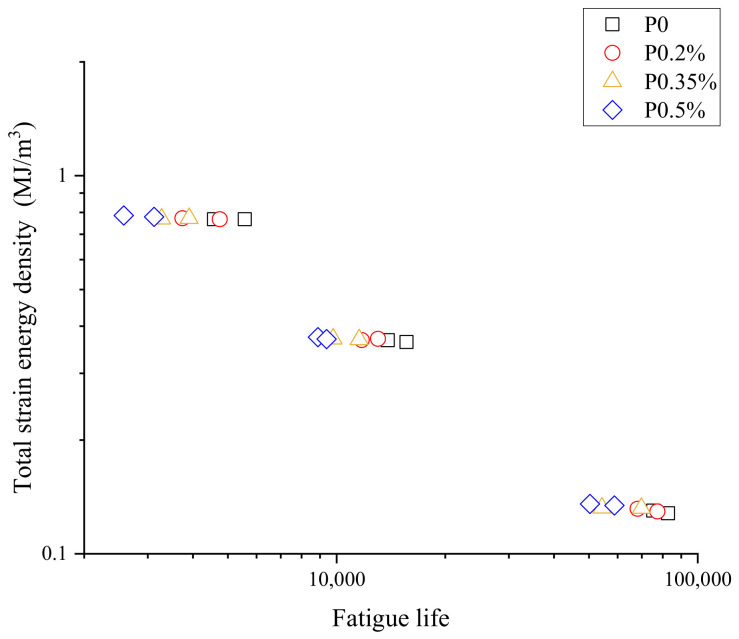
The relationship between the strain energy density and life expectancy.

**Figure 10 materials-15-04558-f010:**
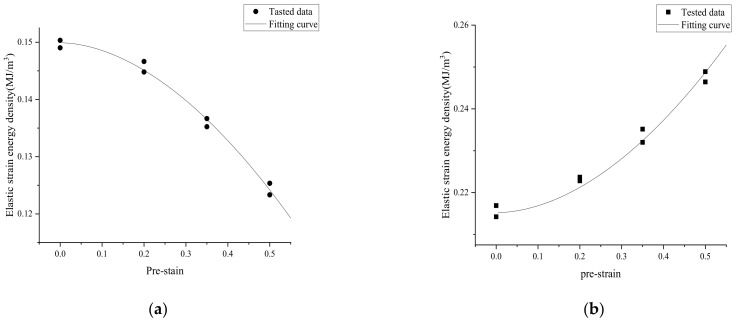
Relationship strain energy density and pre-strain at 0.15% strain amplitude. (**a**) Elastic strain energy density and pre-strain; (**b**) Plastic strain energy density and pre-strain.

**Figure 11 materials-15-04558-f011:**
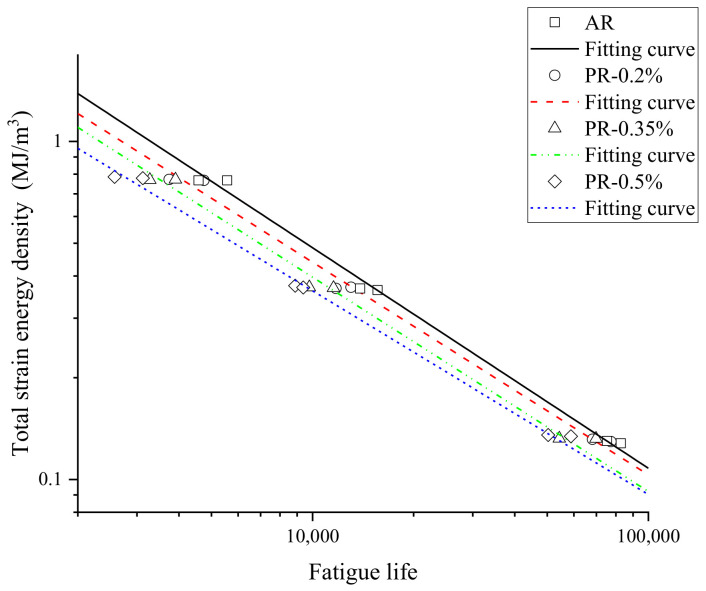
Total strain energy density and fatigue life curve.

**Figure 12 materials-15-04558-f012:**
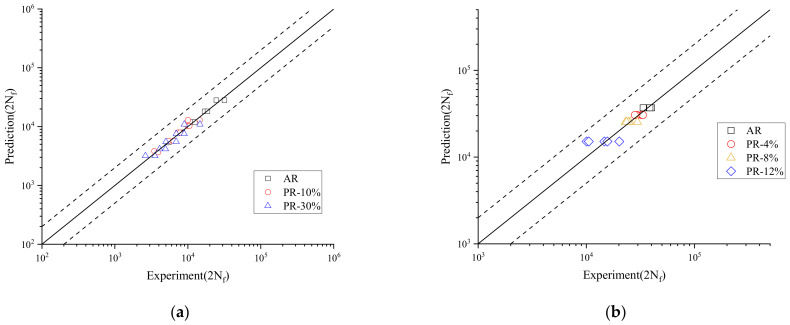
Comparison of fatigue life prediction. (**a**) Material steel Fe-18 Mn; (**b**) Material complex−phase steel CP800.

**Table 1 materials-15-04558-t001:** Main chemical composition (wt%) of high-strength steel Q345.

	C	Si	Mn	P	S	Al	Fe
Q345	0.16	0.30	1.23	0.015	0.003	0.035	Bal.

**Table 2 materials-15-04558-t002:** Mechanical parameters of the welded joint.

Material Properties	Welded Joint	Q345
Elastic modulus (GPa)	205.4	209.5
Yield strength (MPa)	325.1	351.9
Tensile strength (MPa)	440.6	512.8
Poisson’s ratio	0.28	0.29

**Table 3 materials-15-04558-t003:** Strain energy density value versus fatigue life.

Strain Amplitude	Number of Specimens	Elastic Strain Energy Density (MJ/m^3^)	Plastic Strain Energy Density (MJ/m^3^)	Total Strain Energy Density (MJ/m^3^)	Fatigue Life (Cycle)
0.2%	4	0.18757	0.57942	0.766996	4578
	6	0.18603	0.58115	0.767188	5570
	17	0.181211	0.59079	0.772001	3744
	19	0.179952	0.58701	0.766962	4746
	32	0.169569	0.6021	0.771669	3907
	34	0.171722	0.59772	0.769442	3276
	47	0.161759	0.62347	0.785229	2573
	51	0.160656	0.61795	0.778606	3124
0.15%	8	0.149018	0.21422	0.363238	15623
	10	0.150322	0.21689	0.367212	13,863
	22	0.146631	0.22368	0.370311	13,027
	28	0.144796	0.22281	0.367606	11,744
	33	0.13665	0.23201	0.36866	11,548
	38	0.135235	0.23516	0.370395	9780
	45	0.125347	0.24888	0.374227	8879
	55	0.123323	0.24643	0.369753	9380
0.1%	11	0.107484	0.02254	0.130024	75,328
	14	0.106637	0.02135	0.127987	82,798
	29	0.10293	0.02649	0.12942	77,326
	31	0.103823	0.02771	0.131533	68,173
	37	0.097021	0.03502	0.132041	69,813
	44	0.095916	0.03632	0.132236	54,311
	50	0.088879	0.04528	0.134159	58,809
	59	0.08862	0.04679	0.13541	50,282

**Table 4 materials-15-04558-t004:** Fatigue parameters.

$C_{1}\;(\mathrm{MJ}/{\mathrm{mm}}^{3})$	$d_{1}$	$C_{2}\;(\mathrm{MJ}/{\mathrm{mm}}^{3})$	$d_{2}$	$\alpha_{1}$	$\beta_{1}$	$\alpha_{2}$	$\beta_{2}$
−0.219	−3.948	197.3	−0.6524	−183.4	0.305	902.2	3.764

## Data Availability

Not applicable.

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
