# Peer review of "An Energy-Based Method for Lifetime Assessment on High-Strength-Steel Welded Joints under Different Pre-Strain Levels"

_materials, 2022, doi:10.3390/ma15134558_

Round 1

Reviewer 1 Report

Summary

Reviewer comments on “Comparative Investigation of Microstructure, Mechanical Properties and Corrosion of TIG and ATIG Welded 316L Austenitic 3 Stainless Steel”. Manuscript ID “materials-1456725

Upon literature search, it seems that the work is novel and has a good combination of experiment and model development with weak validation. The authors are required to add more thorough references that are key references with data that corroborates their model.

Hence, based on comments, this work seems to be suitable but needs to be improved further.

- - - - - - - - - - - - - - - - - - - - - - - - - - - - - - - - - - - - - - - - - - - - - - - - - - - - - - - - - - - - - - - - - - - - - - -

Major Comments

·       The validation of the model seems not verified by several papers. It has been discussed with reference to one paper which indicated a very weak validation, and this is the reason why there is only one reference in the discussion. The same is represented in the introduction wherein the authors are comparison different trends of pre-strain for aluminum and different materials that do not corroborate in the later parts of the discussion. Please add more data/verification from more papers.

Language

·       The language requires considerate revision. Especially for the past present and future tenses.

·       Replace harmful with another synonym, Intro para 2

Minor Comments

Abstract

·       Please mention the type of steel in the abstract.

Introduction

·       Please improve the flow of literature in the introduction.

·       Literature review is missing, and the work of many authors on this subject matter is missing.

Methodology

·       Figure 2 is not a diagram.

·       Please elaborate on fatigue testing in more detail. Add details about the machine and parameters of testing.

·       Why did the authors select Q345 Steel? Any reason?

Results and Discussion

·       Is experimental verification of data with just 1 result Fe-18 Mn justified to validate your model?

·       The authors claim that this model is valid to describe “The effect of the pre-strain on the fatigue properties of welded joints” but they have not verified with more references, nor they have applied it on other data available, which means that the discussion and comparison along with validation of data with other given data are weak.

Conclusion

·       In the conclusion, the authors mentioned “verified by the experimental data from TWIP steel Fe-18Mn.”, please add the reference here as well.

References

1.     It seems that key references are required to be added further.  

Review conclusion - - - - - - - - - - - -

The authors need to find more justifiable references in the introduction and continue the same flow in their discussion to show that their model applies to a varied sample data and can be generalized as claimed by the authors in different sections of their manuscript. 

Reviewer 2 Report

The paper presents an interesting study on the influence of pre-strain on the lifetime assessment of steel welded joints. The paper is well written, with carefully executed experiments and theoretical modelling whose results are well supported by the experimental results. It is written in fluid English, with just a few spots that need slight corrections:

page 1, line 5 from below: was relation to the loading history – was related to ...

page 2, line 7: it was opposite – the authors probably mean 'decreased'.

page 2, line 1 from below: slight less – slightly less

page 3, line 3 from below: marked – was marked

page 6, line 6 from below: ,.

page 7, eq. (1) The integral sign shall be replaced with that for the closed-path integral .

page 7, line 5 and eq.(2): The tensile positive ...  For a casual reader, this derivation can be made transparent expanding the sentence like: 'The tensile positive elastic strain energy is calculated as [27-31], assuming the linear relation σ=Eε valid in the elastic regime:’

page 9, lines 6-7: with pre-strain and pre-strain – with pre-strain and without it (?)

Round 2

Reviewer 1 Report

The authors did not respond to the following comment. 

The validation of the model seems not verified by several papers. It has been discussed with reference to one paper which indicated a very weak validation, and this is the reason why there is only one reference in the discussion. The same is represented in the introduction wherein the authors are comparison different trends of pre-strain for aluminum and different materials that do not corroborate in the later parts of the discussion. Please add more data/verification from more papers.

I leave it to the editor if this needs to be clarified. The remaining rest of the revisions are okay. 

Author Response

Please have a look at the attachment
